# Foot-and-Mouth Disease Space-Time Clusters and Risk Factors in Cattle and Buffalo in Bangladesh

**DOI:** 10.3390/pathogens9060423

**Published:** 2020-05-29

**Authors:** A K M Anisur Rahman, SK Shaheenur Islam, Md. Abu Sufian, Md. Hasanuzzaman Talukder, Michael P. Ward, Beatriz Martínez-López

**Affiliations:** 1Department of Medicine, Bangladesh Agricultural University, Mymensingh 2202, Bangladesh; 2Department of Livestock Services, Krishi Khamar Sarak, Farmgate, Dhaka 1215, Bangladesh; s_islam73@live.com (S.S.I.); sufian04@yahoo.com (M.A.S.); 3Department of Parasitology, Bangladesh Agricultural University, Mymensingh 2202, Bangladesh; talukdermhasan@bau.edu.bd; 4Sydney School of Veterinary Science, The University of Sydney, Camden 2570, NSW, Australia; michael.ward@sydney.edu.au; 5Department of Medicine and Epidemiology, Center for Animal Disease Modeling and Surveillance, School of Veterinary Medicine, University of California, Davis, CA 95616, USA; beamartinezlopez@ucdavis.edu

**Keywords:** FMD, cluster analysis, risk mapping, risk-based surveillance

## Abstract

Foot-and-mouth disease (FMD) is highly endemic in Bangladesh. Using passive surveillance data (case records from all 64 districts of Bangladesh, 2014–2017) and district domestic ruminant population estimates, we calculated FMD cumulative incidence per 10,000 animals at risk per district, conducted cluster (Moran’s spatial autocorrelation and scan statistics) and hotspot analysis (local indicator of spatial association statistic), created predictive maps and identified risk factors using a geographically weighted regression model. A total of 548,817 FMD cases in cattle and buffalo were reported during the four-year study period. The highest proportion (31.5%) of cases were reported during the post-monsoon season, and from Chattogram (29.2%) division. Five space-time clusters, 9 local clusters, and 14 hotspots were identified. Overall, higher cumulative incidences of FMD were consistently predicted in eastern parts of Bangladesh. The precipitation in the pre-monsoon season (*p* = 0.0008) was positively associated with FMD in Bangladesh. Results suggest climate plays an important role in the epidemiology of FMD in Bangladesh, and high risk zones exist. In a resource limited-setting, hotspots and clusters should be prioritized for vaccination coverage, and surveillance for FMD should be targeted in eastern areas of Bangladesh and during the post-monsoon season.

## 1. Introduction

Foot-and-mouth disease (FMD) is an extremely contagious and acute viral disease of domestic and wild cloven-hoofed animals. It is highly endemic in Africa and Asia including Bangladesh [1]. The most important manifestations of FMD are fever, lameness, drooling, and vesicular lesions of the tongue, feet, and teats [2]. Vesicular fluid, vesicular epithelium, respiratory aerosols, and droplets, milk, urine, and semen of acutely infected animals contain large amounts of FMD virus. Susceptible animals are infected through direct contact with respiratory aerosols and droplets from acutely infected animals, or indirectly via the environment or mechanically by persons, vehicles, wild animals, or birds [3]. FMD causes reduction in milk yield (up to 33%), reduced growth, mortality in calves, loss of traction power, and abortion [4]. In endemic countries, an economic loss of between US $6.5 and 21 billion occurs every year [4].

A large FMD-susceptible livestock population exists in Bangladesh: 24 million cattle, 26 million goats, 3.5 million sheep, and 1.5 million buffaloes [5]. About 70% (20% directly and 50% indirectly) of rural people in Bangladesh rely on livestock production. The contribution of livestock to the national economy was 1.47% in 2018–2019 and it is growing at 3.47% annually [5]. However, diseases are one of the major constraints to the advancement of the livestock industry in Bangladesh. FMD is considered to be the most economically important disease of livestock in Bangladesh, with annual losses estimated to be about US $125 million [6]. A mortality of 51% in calves and 66.6% reduction of milk yield have also been reported in some areas of the country [3,7]. Every year several outbreaks of FMD occur in different districts of Bangladesh, particularly during winter and pre-monsoon season [3,7]. FMD morbidity was reported to be significantly higher in cattle (36%) than buffaloes (23.3%) and sheep/goats (4.8%), and FMD prevalence to be significantly higher in male than female and older than younger cattle [8,9]. Cross border (informal and formal) and uncontrolled within-country animal movement are important risk factors for FMD transmission in Bangladesh [6,10,11]. Pigs act as an amplifying host [4] for FMD virus and nomadic pig herds are also considered another risk factor for FMD [6]. However, there are approximately eight million pigs in Bangladesh [12] and their distribution is more in northern and hilly districts of Bangladesh.

Three serotypes of FMD viruses have been reported: O, A, and Asia-1; among them, serotype O is most prevalent in Bangladesh [11,13]. These serotypes are immunologically distinct and hence vaccines prepared from one serotype do not protect against others [14]. FMD vaccines are available (~3 million doses) in Bangladesh but immunity is short-lived, a maximum of 6 months [14,15]. There is no approved strategy for FMD control in Bangladesh. Only conscientious farmers use vaccine routinely in crossbred cattle against FMDV in cattle dense districts of Bangladesh, and that covers 10–15% of total susceptible cattle population. Moreover, Department of Livestock Services (DLS) is doing emergency vaccination (ring vaccination) at outbreak sites [5]. The current status of FMD vaccination coverage for buffalo in Bangladesh is not known. However, one study performed in Bagerhat district in 2013 suggested that 56.7% farmers use FMD vaccine for their buffalo [16]. This finding might not reflect the vaccine coverage of other buffalo rearing areas in Bangladesh due to non-random sampling.

FMD-free status enables meat exports, but it might be difficult for Bangladesh to gain national FMD-free status. However, even if some FMD-free areas can be created it will be helpful for meat exports [17]. Risk-based strategic vaccination could be of great value to achieve this goal rather than mass vaccination in a resource-limited setting such as Bangladesh [18]. Knowledge of disease high risk areas (i.e., hotspots) and better characterization of the risk factors will enable policy-makers to initiate targeted, more cost-efficient, control measures in Bangladesh. However, there is no published report on FMD high-risk areas or spatio-temporal risk factors in Bangladesh. Hence, our aims in this study were to describe FMD time-space clusters, determine risk factors, and develop predictive FMD risk maps using four years of passive surveillance data at the national level.

## 2. Results

### 2.1. Descriptive Statistics

The median (interquartile range, IQR) of four years (2014–2017) average cattle and buffalo population per district was 416,370 (317,260–554,624). A total of 548,817 FMD cases were reported from 64 districts during the four years period (Table 1). The overall median (IQR) cumulative incidence of FMD per ten thousand cattle and buffalo per district in Bangladesh was 135 (86–241) during the four-year study period. The total number of FMD cases were 153,421 in 2014; 102,767 in 2015; 140,270 in 2016 and 152,359 in 2017. The median (IQR) number of cases of FMD per district were 1726 (1010–3276) in 2014; 851 (406–1880) in 2015; 1407 (576–2997) in 2016; and 1376 (565–2642) in 2017. The highest number of FMD cases were reported in the month of November (11.1%) and during the post-monsoon season (31.5%). The proportion of reported cases was highest in Chattogram division (29.2%) and lowest in Mymensingh division (1.2%) (Table 1).

### 2.2. Clustering and Risk Prediction

The estimated Moran’s I were 0.16 (Z-score = 2.15, P=0.03), 0.26 (Z-score = 3.13, *p* = 0.002), 0.12 (Z-score = 1.94, *p* = 0.05) and 0.18 (Z-score = 2.22, *p* = 0.01), respectively, in 2014, 2015, 2016 and 2017, indicating strong global clustering of FMD cases. FMD hotspots, high-high, low-low clusters, and predicted cumulative incidence per 10,000 populations are shown in Figure 1, Figure 2, Figure 3, Figure 4, and Figure 5 for 2014, 2015, 2016, 2017 and for all four years together, respectively. High-high (high numbers of cases in a district with high numbers of cases in surrounding districts) local FMD clusters were observed in 9 districts: Noakhali, Feni, Laskmipur, Chandpur, Dhaka, Narayanganj, Narsingdi, Khagrachari, and Rajshahi (Figure 1b, Figure 2b, Figure 3b, Figure 4b, and Figure 5b). Among them, Laskmipur, Noakhali and Feni districts showed high-high clusters frequently (Figure 2b, Figure 3b, Figure 4b, and Figure 5b). Low-low (low numbers of cases in a district with low numbers of cases in surrounding districts) FMD clusters were observed in 10 districts which centered mostly in two divisions in Bangladesh: Sherpur, Jamalpur, Mymensingh (Mymensingh division), Khulna, Satkhira, Bagerhat, Jashore, Magura (Khulna division), Barguna, and Nilphamari (Figure 1b, Figure 2b, Figure 3b, Figure 4b, and Figure 5b). Some high-low (high numbers of cases in a district with low numbers of cases in surrounding districts) and low-high (low numbers of cases in a district with high numbers of cases in surrounding districts) outliers were also detected in 2014 (one high-low), in 2015 (two low-high), in 2016 (two high-low and one low-high), in 2017 (two high-low) and in all four years together (one high-low and two low-high). FMD hotspots (an area where FMD incidence is higher compared with the surrounding area) were detected in 14 districts: Cumilla, Feni, Laskmipur, Noakhali, Chattogram, Cox’s Bazar, Sylhet, Sunamganj, Narayanganj, Gazipur, Natore, Jhalokati, Meherpur and Thakurgaon. The most frequently encountered FMD hotspots were Cumilla, Feni, Sylhet, and Narayanganj districts (Figure 1a, Figure 2a, Figure 3a, Figure 4a, and Figure 5a).

Five space-time clusters in different locations were identified using scan statistics, including 20 districts in five divisions. Three out of five clusters were detected in 2014. Most clusters occurred between June to December (Table 2). The clusters are shown in Figure 6b.

The empirical Bayesian kriging (EBK) selected used an exponential semivariogram model with empirical transformation. Both the subset size and number of simulations were 100. The EBK searched a standard circular neighborhood for each observed location that included a minimum and maximum of 10 and 15 neighbors, with a radius of 1.1. The map of predicted FMD cumulative incidence had similarities with the hotspot and cluster maps in each of the respective years. Higher cumulative incidences of FMD were predicted mostly in north-east, east and south-east parts of Bangladesh in each year and all four years together. In contrast, the lower cumulative incidences of FMD were predicted in north and south-west areas of Bangladesh in each year and all four years together (Figure 1c, Figure 2c, Figure 3c, Figure 4c, and Figure 5c).

### 2.3. Spatial Risk Factors

The ordinary least squares (OLS) regression assumption of normality of the error was accepted as reflected by the non-significant Jarque-Bera test result. The Breusch–Pagan test result was not significant indicating constant variance of the dependent variables around the regression line. As expected, Moran’s I test result was highly significant (*p* < 0.001) indicating spatial dependence. Based on the results of OLS regression, a spatial-error model was chosen for univariable and multivariable spatial regression analyses. In univariable geographically weighted regression (GWR) analysis, the average temperature during the monsoon, precipitation in the pre-monsoon season and water bodies land surface were significantly associated with FMD in Bangladesh (Table 3) and selected as potential good candidate predictors to be included in the final multivariable model. In the final multivariable GWR model, only precipitation in the pre-monsoon season (Estimate = 0.0033, SE = 0.0010, *p* = 0.02) was positively associated with FMD in cattle and buffalo in Bangladesh. The final model had a R-squared value of 0.279. Figure 6 shows maps of cattle and buffalo population, significant space-time clusters for FMD, and total number of FMD cases in cattle and buffalo in Bangladesh. The highest precipitation (187–322 mm) was recorded in four districts (Meherpur, Sunamganj, Sylhet and Habiganj). The maps of risk factor, predicted error, and predicted FMD cumulative incidence in cattle and buffalo in Bangladesh are shown in Figure 7. The highest predicted cumulative incidence of FMD and the highest precipitation in the pre-monsoon season were observed in Sylhet, Sunamganj, Habiganj and Meherpur districts.

## 3. Discussion

Using four years of national-level passive surveillance data we have described for the first-time the hotspots, clusters in space and time, and climatic risk for FMD in cattle and buffalo in Bangladesh. Some high-high and low-low clusters, and hotspots were identified, mostly in eastern Bangladesh, and the distribution of FMD in cattle and buffalo in Bangladesh was found to be associated with precipitation in the pre-monsoon season. The results suggest that risk-based vaccination strategies can significantly help to control FMD in Bangladesh.

About 5.5 million FMD cases were reported in cattle and buffalo in Bangladesh during the four-year study period. As all farmers do not have access to upazila (sub-district) veterinary hospitals, the total number of FMD cases in cattle and buffalo is likely much higher than that reported here. The highest number of cases was reported in the month of November and in Chattogram division. A similar finding—based on descriptive analysis only—was reported by another retrospective epidemiologic study in Bangladesh [15]. The average temperature (23.5 °C) and relative humidity (RH) (66%) in November are favorable for the survival of FMDV, which might be a reason for the higher prevalence of FMD [19]. Chattogram division shares a common boundary with India and Myanmar. Extensive cross-border animal movement from India, Nepal, and Myanmar to Bangladesh is reported to be an important pathway of FMDV spread. Molecular evidence of the FMDV serotypes in Bangladesh, India, Nepal, and Myanmar also suggest transboundary transmission [10,11,17]. FMD is a vaccine-preventable disease. Bangladesh is trying constant to boost up national vaccine production. At present, approximately 3 million doses of trivalent vaccines are produced for cattle immunization; however, this amount is not sufficient to cover completely susceptible cattle population. In this regard, some imported polyvalent vaccines like Aftovaxpur^®^ (Boehringer-Ingelheim, Ingelheim am Rhein, Germany) and RAKSHA (Indian Immunologicals Ltd., Hyderabad, India) are also available in Bangladesh, but their price is higher than locally available vaccine. However, the level of protection in animal is not satisfactory against the circulating FMDVs. Inappropriate vaccination in combination with uncontrolled animal movement and lack of awareness among stakeholders on FMDV transmission dynamics are also the pivotal determinants for recurrent outbreaks in Bangladesh [13].

In this study, we recommend high-high clusters and hotspot districts be prioritized for FMD vaccination. Cumilla, Noakhali, Feni, Laskmipur, Chandpur, Chattogram, Cox’s Bazar and Khagrachari districts of Chattogram division; Dhaka, Narayanganj, Gazipur, and Narsingdi districts of Dhaka division; Natore and Rajshahi districts of Rajshahi division; Meherpur district of Khulna division; Thakurgaon district of Rangpur division; Jhalokati district of Barishal division and Sylhet and Sunamganj districts of Sylhet division should be prioritized for vaccination coverage. To prevent FMD spread ring vaccination can be carried out around known hotspots and high-high clusters [20]. Moreover, cattle and buffaloes in these districts should be vaccinated first, followed by other species (goats, sheep, and pigs).

FMD endemic countries suffer from trade restrictions [21]. In a resource-limited setting such as Bangladesh, it might be expensive to gain FMD free status. However, interestingly some low-low FMD clusters were detected in our study which might be potential areas for developing FMD free status. For this, we recommend the inclusion of Mymensingh (Sherpur, Jamalpur, Mymensingh) and Khulna divisions (Khulna, Satkhira, Bagerhat, Jashore, Magura). However, the livestock movement network in the country needs to be well-documented first so that check points can be established for better tracing and monitoring of FMD outbreaks. District and sub-district livestock officers, traders, and farmers should be motivated and involved in this process. Initially, a buffer zone containing vaccinated animals may be used to separate low-low risk areas from other areas [22]. Strict control of animal movement from high risk areas to low risk areas, strong biosecurity practices, and introduction of healthy animals to herds will help to achieve an FMD free zone.

Although more FMD cases were reported in the post-monsoon season, pre-monsoon precipitation was identified as a spatial risk factor for the cumulative incidence of FMD. The survival of FMDV is dependent on various climatic and microclimatic conditions. The optimal conditions for FMDV survival in the environment are <20 ℃ temperature and >55% relative humidity (RH) [19]. Temperature <20 ℃ only occurs in the winter season in Bangladesh. The average temperature in the pre-monsoon season in Bangladesh is 27.4 ℃ (relatively high) but the RH is 57.3% (favorable). As temperatures increase, desiccation reduces viral survival; however, high RH will reduce the potential for desiccation and increase the survival rate of virus [19]. It was also reported from one study that the prevalence of FMD is significantly higher in sub-humid than semi-arid zones [23]. Another study investigated 930 FMD outbreaks during 1988 to 1991 in Bangladesh and reported a significantly higher occurrence of FMD in pre-monsoon and winter seasons [7]. Meherpur, Sunamganj, Sylhet, and Habiganj districts have the highest precipitation records and therefore should be prioritized for FMD control.

Our study has some limitations. There might have been reporting bias as FMD cases are mostly reported from areas that surround upazila veterinary hospitals. There is no active surveillance system for FMD and limited control program for strategic vaccination. In this scenario, the passive surveillance data is also valuable at least to have some idea about hotspots and clusters that can help control decisions. FMD is mostly diagnosed by clinical signs and infrequently by laboratory confirmation. However, field veterinarians know this disease very well and the characteristic clinical signs of FMD in cattle and buffalo rarely creates confusion with other diseases endemic in Bangladesh. In addition, to increase the positive predictive value of the presumptive diagnosis, we included only those cases which presented with all clinical signs (fever, oral lesion, lameness, and salivation), a very specific case definition. The data contained very limited information (date and number of cases) and some monthly reports were missing in some districts. The distribution of FMD cases according to species, breed, age and gender were not available, although this information is usually collected for each case before being aggregated. The data for number of cases per herd was not available. Since cases were aggregated, we argue that there was no systemic bias because we have no reason to suspect that more cases are likely to be diagnosed in some herds (e.g., small herds) that other herds (e.g., large herds). Therefore, between districts there should not be bias caused by this aggregation. Within-country and transboundary animal movement data, pig population in every district, data on vaccination coverage in each district, and live animal market data were not available; these are all important factors in the epidemiology of FMD. Certainly, analyzing properly collected passive surveillance data can generate valuable knowledge about diseases, which is important for disease control decisions. DLS do not have enough manpower to collect animal health and disease information at the farm level. However, the improvement of disease data collection at upazila level would be a valuable asset in terms of exploring the epidemiology of endemic diseases.

## 4. Materials and Methods

### 4.1. Data

#### 4.1.1. FMD Case Data

Bangladesh has 8 divisions, 64 districts, and 491 sub-districts (or upazilas). Cases (single cattle or buffalo) attending to the upazila veterinary hospitals are recorded routinely as a part of disease surveillance. Sub-districts report monthly surveillance data to districts, and districts to divisions, which compile and forward data to the Epidemiology Unit of the Department of Livestock Service (DLS). FMD cases in cattle and buffalo during 2014 to 2017 were obtained from DLS and used for this study. Field veterinarians are capable of making a correct diagnosis of FMD based on history and clinical signs alone. However, to increase the positive predictive value of the presumptive diagnosis of FMD we only included those records in which the animals showed all clinical signs that characterize FMD that is fever, oral lesions, excessive salivation, and lameness [24]. Except FMD, there is no disease of cattle and buffalo in Bangladesh, which has all of the signs together. Nine-field disease investigation laboratories assist in the tentative diagnosis, and two national-level laboratories (Bangladesh Livestock Research Institute and Central Disease Investigation Laboratory) provide confirmatory diagnosis of FMD through advanced techniques (ELISA and RT-PCR). About 1–2%, passive surveillance cases were confirmed by testing clinical samples (tongue epithelial tissue, vesicular fluid) through advanced techniques. The data included information on case date and district name. Data on species involved and their demographic characteristics were not available. The number of species of cattle and buffalo in each district was also available from DLS.

#### 4.1.2. Environmental and Climatic Data 

We downloaded average monthly temperature (°C × ~1 km^2^), precipitation (mm × ~1 km^2^), solar radiation (KJm^−2^ day^−1^ × ~1 km^2^) and wind speed (m s^−1^ × ~1 km^2^) from world climate data (www.worldclim.org). The Bangladesh district specific monthly mean temperature, precipitation, solar radiation and wind speed records were extracted from corresponding raster files using Zonal statistics as Table option (Spatial Analyst Tools, ArcGIS 10.7.1, Environmental System Research Institute, Redlands, CA, USA). These monthly records were aggregated into four seasons: winter (December to February), pre-monsoon (March to May), monsoon (June to August) and post-monsoon (September to November). The length of road, railroad and river, country mask elevation and land-cover data were downloaded from the DIVA-GIS website (www.diva-gis.org). These elevation and land-cover data were read, and Bangladesh district-level values were calculated in ArcGIS 10.7.1 using a spatial join. The length of road, railroad and river (polyline) and area of water bodies (polygon) per district were calculated by adding a respective field in the attribute table. The Bangladesh district shapefile containing log-transformed total number of FMD cases, road, river and railroad length, elevation, land-cover, area of water bodies, seasonal temperature, precipitation, wind speed, and solar radiation data were used for the spatial regression analysis

### 4.2. Analysis

#### 4.2.1. Descriptive Statistics

The FMD case database was integrated into R-language and data were aggregated and summarized by month-, season-, and year-specific using the “aggregate” and “summary” function in the R 3.6.1 [25] “stats” and “base” packages, respectively.

#### 4.2.2. Spatial Analysis

##### Cluster Analyses

The district cumulative incidence of FMD per 10,000 cattle and buffalo population was used for spatial analysis (Appendix A). First, we used a global cluster method (Moran’s I) [26] to identify if clustering was present in the study region. Then we explored where the clusters were located using three approaches: local indicators of spatial association (LISA) [27], Getis Ord [28] and SaTScan [29]. The rationale for performing three different clustering methods are that LISA detects outliers efficiently [30], Getis Ord identifies locations surrounded by a cluster of high or low values and SaTScan detects small, compact clusters both in space and time [31]. Because some methods can lack power or detect different disease patterns, use of a range of methods when investigating spatial [32] and temporo-spatial [33] disease patterns have been advocated.

The space-time cluster analysis was performed using SaTScan Version 9.6 (http://www.satscan.org). We used a retrospective Poisson model to search, detect and test for significance of FMD space-time clusters [29] in cattle and buffalo in Bangladesh. The search was performed using circular spatial moving windows up to 10% of the population at-risk and temporal windows up to 6 months duration. A cluster in space and time was defined when there were more cases observed (*O*) within the scanning window than expected (*E*), and its statistical significance was evaluated using the log likelihood ratio statistic. The corresponding *p*-value was obtained via Monte Carlo simulations (*n*  =  999). Significant clusters were visualized using ArcGIS 10.7.1 (Environmental System Research Institute, Redlands, CA, USA).

For geostatistical prediction and generation of risk maps, we used the EBK method (Geostatistical Analyst, Geostatistical Wizard, ArcGIS 10.7.1, Environmental System Research Institute, USA). We created risk maps for each year to predict FMD cases at unmeasured locations using the observed cases at surrounding locations [34]. The EBK method automatically calculates model parameters through a process of subsetting and simulations. 

##### Geographically Weighted Regression (GWR)

A Bangladesh district shapefile containing all data was imported into GeoDa 1.14.0 [35]. We created a spatial weight file of districts, based on Euclidean distance between district centroids. Initially, an ordinary least square regression was fit to the log-transformed district FMD cumulative incidence as the dependent variable in the model. The normality of error was tested by the Jarque-Bera test [35]. Homoscedasticity of random coefficients and spatial dependence were tested respectively by Breusch–Pagan test and Moran’s I test [35].

The results of the ordinary least square regression analysis lead us to choose univariable and multivariable GWR models. Variables significant at 10% level in univariable analysis were used for multivariable analysis. A manual forward multivariable GWR analysis was conducted to identify risk factors for FMD. The best univariate model was selected based on the lowest Akaike’s information criterion (AIC) value. Then the remaining variables were added in turn, based on AIC. The final model selected had the lowest AIC.

## 5. Conclusions

We have advanced the knowledge on the epidemiology of FMD in Bangladesh, identified spatio-temporal clusters (hotspots, high-high, low-low clusters) and the influence of climate. Hotspots and high-high clusters should be selected for vaccination and low-low clusters can provide the foundations for creating FMD-free zones in Bangladesh. Cattle and buffaloes within hotspots and high-high cluster districts should be prioritized for vaccination, followed by other species (goats, sheep, and pigs). Additional protective factors associated with low-low cluster areas and risk factors for high-high clusters and hotspots should be explored. The DLS passive surveillance network could be utilized more efficiently if some additional information is included in the report (e.g., address/geographic coordinates of the farm, herd size, herd composition, vaccination status, number of animals vaccinated) and if individual records for each case with information about animal species, age, sex, breed, and other animal characteristics are provided. To achieve the above targets and goals, an effective roadmap strategy yet to be approved and implemented as a part of FAO- and OIE-proposed Progressive Control Pathway for FMD (PCP-FMD) is much demanding.

## Figures and Tables

**Figure 1 pathogens-09-00423-f001:**
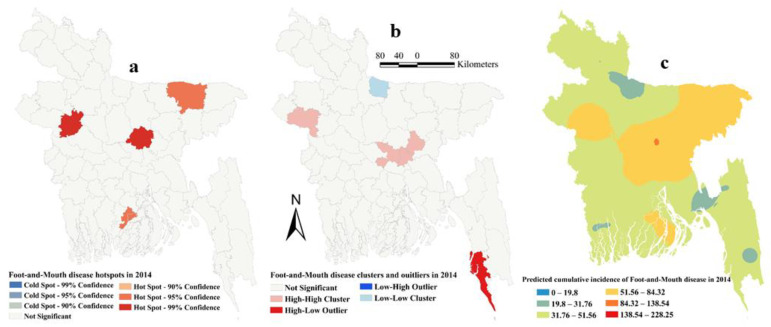
Foot-and-mouth disease hotspots, clusters and outliers, and risk map in 2014 (**a**) Hotspots; (**b**) Clusters and outliers; (**c**) Risk map.

**Figure 2 pathogens-09-00423-f002:**
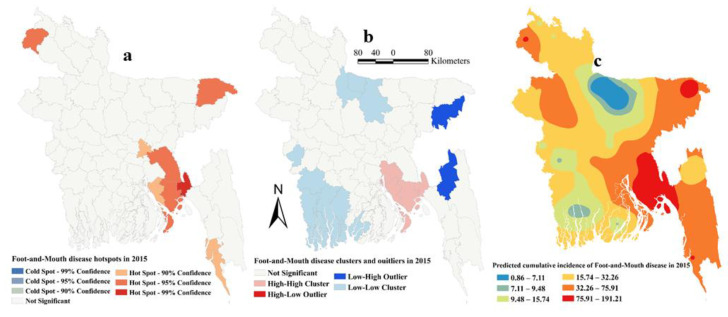
Foot-and-mouth disease hotspots, clusters and outliers, and risk map in 2015 (**a**) Hotspots; (**b**) Clusters and outliers; (**c**) Risk map.

**Figure 3 pathogens-09-00423-f003:**
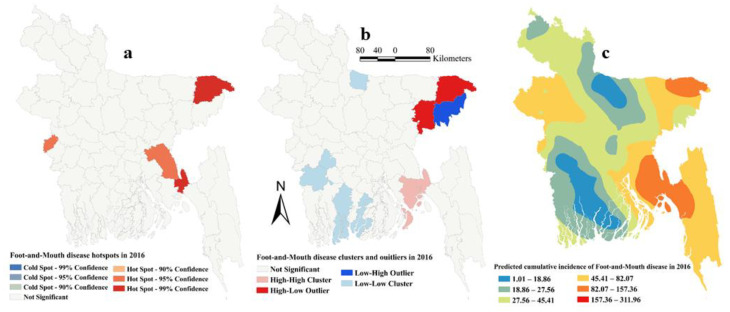
Foot-and-mouth disease hotspots, clusters and outliers, and risk map in 2016 (**a**) Hotspots; (**b**) Clusters and outliers; (**c**) Risk map.

**Figure 4 pathogens-09-00423-f004:**
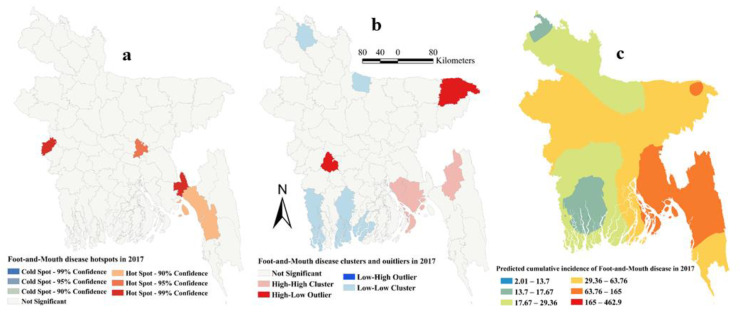
Foot-and-mouth disease hotspots, clusters and outliers, and risk map in 2017 (**a**) Hotspots; (**b**) Clusters and outliers; (**c**) Risk map.

**Figure 5 pathogens-09-00423-f005:**
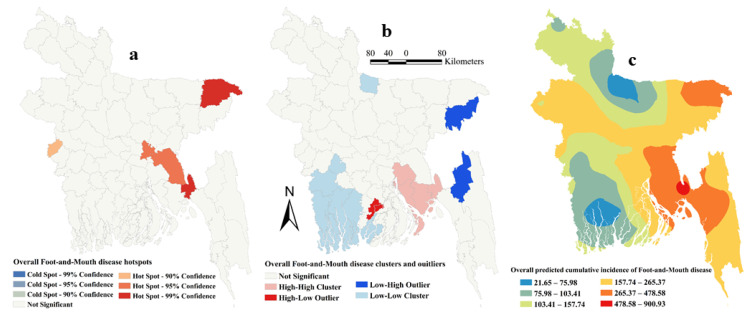
Foot-and-mouth disease hotspots, clusters and outliers. and risk map based on overall cumulative incidence (**a**) Hotspots; (**b**) Clusters and outliers; (**c**) Risk map.

**Figure 6 pathogens-09-00423-f006:**
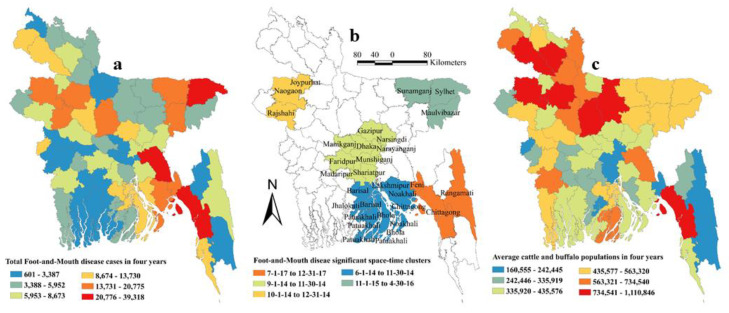
Map of Bangladesh showing average population of cattle and buffalo, significant space-time cluster of foot-and-mouth disease (FMD) and total number of FMD cases in four years (**a**) Average population of cattle and buffalo in four years; (**b**) Significant space-time clusters; (**c**) Total number of FMD cases in four years.

**Figure 7 pathogens-09-00423-f007:**
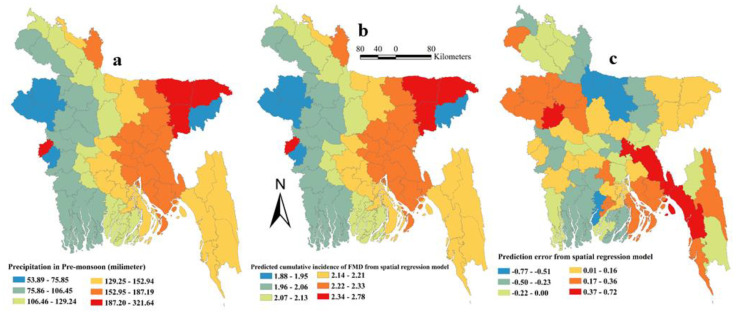
Map of Bangladesh significant spatial risk factor, predicted cumulative incidence of Foot-and-mouth (FMD) disease and prediction error from the geographically weighted regression model (**a**) Precipitation in Pre-Monsoon (millimeter); (**b**) Predicted cumulative incidence of FMD from geographically weighted regression model; (**c**) Prediction error from geographically weighted regression model.

**Table 1 pathogens-09-00423-t001:** Distribution of foot-and-mouth disease in cattle and buffalo based on passive surveillance data reported from 64 districts in Bangladesh during 2014–2017.

Variable	Cases (Average Cumulative Incidence per 10,000 Population)	% (95% Confidence Interval)
Year		
2014	153,421 (54.5)	27.9 (27.8–28.1)
2015	102,767 (36.10)	18.7 (18.6–18.8)
2016	140,270 (48.2)	25.6 (25.4–25.7)
2017	152,359 (54.2)	27.7 (27.6–27.9)
**Month**		
December	46,752	8.5 (8.4–8.6)
January	36,942	6.7 (6.6–6.8)
February	39,560	7.2 (7.1–7.3)
Winter (December–February)	123,254	22.5 (22.3–22.6)
March	36,146	6.6 (6.5–6.7)
April	40,418	7.3 (7.2–7.4)
May	37,459	6.8 (6.7–6.9)
Pre-monsoon (March–May)	114,023	20.8 (20.7–20.9)
June	44,954	8.2 (8.1–8.3)
July	44,284	8.0 (7.9–8.1)
August	49,433	9.0 (8.9–9.1)
Monsoon (June–August)	138,671	25.3 (25.2–25.4)
September	57,187	10.4 (10.3–10.5)
October	54,720	9.9 (9.8–10.0)
November	60,962	11.1 (11.0–11.2)
Post-monsoon (September–November)	172,869	31.5 (31.4–31.6)
**Division**		
Barishal	32,025 (13.8)	5.8 (5.7–5.9)
Chattogram	160,259 (65.6)	29.2 (29.0–29.3)
Dhaka	87,091 (36.9)	15.9 (15.7–15.9)
Khulna	30,528 (11.7)	5.6 (5.5–5.6)
Mymensingh	6,623 (3.9)	1.2 (1.1–1.2)
Rajshahi	90,835 (35.4)	16.6 (16.5–16.7)
Rangpur	53,698 (22.5)	9.8 (9.7–9.9)
Sylhet	87,758 (45.4)	15.9 (15.8–16.1)
Total	548,817	100

**Table 2 pathogens-09-00423-t002:** Significant space-time clusters of foot-and-mouth disease cases reported from 64 districts in Bangladesh between January 2014 and December 2017.

Districts	Radius (Km)	O/E	LLR	Time Period	*p*-Value
Chattogram, Rangamati, Feni	75.7	7.1	24,912.5	7/1/2017–2/31/2017	<0.001
Sylhet, Maulvibazar, Sunamganj	64.8	5.4	13,726.5	11/1/2015–4/30/2016	<0.001
Dhaka, Manikganj, Narayanganj, Munshiganj, Gazipur, Faridpur, Narsingdi, Shariatpur, Madaripur	63.7	3.4	5513.8	9 /1/2014–11/30/2014	<0.001
Bhola, Patuakhali, Noakhali, Lakshmipur, Jhalokati, Barishal	65.4	2.3	3446.4	6/1/2014–11/30/2014	<0.001
Naogaon, Joypurhat, Rajshahi	49.2	2.7	2141.9	10/1/2014–12/31/2014	<0.001

Maximum spatial window = 10% of study area; Maximum temporal window = 6 months; LLR = Log likelihood ratio; O/E = observed/expected.

**Table 3 pathogens-09-00423-t003:** Explanatory variables associated with log-transformed overall cumulative incidence of foot-and-mouth disease in cattle and buffalo in Bangladesh using univariable geographically weighted regression analysis.

Variables	Categories	Coefficient	SE	*p*-Value
Solar radiation (KJ m^−2^)	Winter	0.0001	0.0001	0.4450
	Pre-monsoon	−0.00004	0.0001	0.7429
	Monsoon	−0.00008	0.0002	0.6049
	Post-monsoon	0.0004	0.0003	0.1109
Temperature (℃)	Winter	0.0603	0.0865	0.4858
	Pre-monsoon	−0.1127	0.0759	0.1374
	Monsoon	−0.1834	0.0892	0.0398
	Post-monsoon	0.0578	0.1189	0.6273
Precipitation (mm)	Winter	0.0015	0.0180	0.9301
	Pre-monsoon	0.0034	0.0010	0.0008
	Monsoon	0.0007	0.0004	0.1263
	Post-monsoon	0.0028	0,0021	0.1853
Wind speed (m s^−1^)	Winter	−0.0065	0.2153	0.9758
	Pre-monsoon	−0.0059	0.1392	0.8526
	Monsoon	−0.0202	0.1166	0.8625
	Post-monsoon	−0.0143	0.1841	0.9381
Elevation (m)	-	−0.0016	0.0013	0.2100
Length of river (km)	-	−0.0002	0.0001	0.3227
Area of inland water bodies (km^−2^)	-	−0.00006	0.00008	0.4345
Length of road (km)	-	0.00007	0.0002	0.7548
Length of rail road (km)	-	−0.0003	0.0005	0.6384
Land cover	Tree-cover, broadleaved, evergreen	0.0012	0.0011	0.2774
	Tree-cover, broadleaved, deciduous, closed	0.0028	0.0021	0.1954
	Tree-cover, regularly flooded, saline water	−0.0002	0.0001	0.1517
	Mosaic: tree-cover and other natural vegetation	0.00009	0.0002	0.5511
	Tree-cover, burnt	0.0000005	0.00005	0.9926
	Shrub-cover, closed-open, evergreen	0.0006	0.0003	0.0616
	Shrub-cover, closed-open, deciduous	−0.2111	0.1471	0.1512
	Cultivated and managed areas	−0.00001	0.00003	0.7029
	Mosaic: cropland, tree-cover, other natural vegetation	0.00006	0.0001	0.9533
	Artificial surface and associated areas	0.0002	0.0008	0.7928
	Water bodies	0.0005	0.0002	0.0385

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
