# Peer review of "Foot-and-Mouth Disease Space-Time Clusters and Risk Factors in Cattle and Buffalo in Bangladesh"

_pathogens, 2020, doi:10.3390/pathogens9060423_

Round 1

Reviewer 1 Report

I have only a few minor comments:

  1. You should not write Foot-and-Mouth disease throughout the whole text, it is foot-and-mouth disease
  2. FMD is not endemic in Latin America; the last most recent outbreak was in 2018 in Colombia
  3. Abbreviations, such as DLS, EBK are introduced in the Materials & Methods section but appear in the text much earlier. Please introduce the abbreviations (additionally) at first appearance in the text. Same is true for the term "upazilas". Please, just add a short sub-clause of description to explain that term at its first appearance
  4. The legends of the figures are difficult to read and should be enlarged as well as the resolution
  5. There are quite often additional space characters between words, which was quite eye-catching. Please go through the manuscript and delete excessive space characters
  6. Table 3: I did not understand the meaning of the O/E values. Are these observed or expected values? As it is declared in the headline, I would expect values presented like X/Y. Generally, I did not understand this value. Is there a unit such as number of cases?

Author Response

Point 1: You should not write Foot-and-Mouth disease throughout the whole text, it is foot-and-mouth disease

Response 1: Foot-and-mouth disease changed to foot-and-mouth disease

Point 2: FMD is not endemic in Latin America; the last most recent outbreak was in 2018 in Colombia

Response 2: Latin America deleted from Introduction.

Point 3: Abbreviations, such as DLS, EBK are introduced in the Materials & Methods section but appear in the text much earlier. Please introduce the abbreviations (additionally) at first appearance in the text. Same is true for the term "upazilas". Please, just add a short sub-clause of description to explain that term at its first appearance

Response 3: Now DLS (page 2), EBK (page 7) and upazilas (page 9) were defined at their first appearance.

Point 4: The legends of the figures are difficult to read and should be enlarged as well as the resolution

Response 4: The resolution and font size of legends for all figures have now been increased.

Point 5: There are quite often additional space characters between words, which was quite eye-catching. Please go through the manuscript and delete excessive space characters

Response 5: Additional space between words have been deleted in the revised manuscript.

Point 6: Table 3: I did not understand the meaning of the O/E values. Are these observed or expected values? As it is declared in the headline, I would expect values presented like X/Y. Generally, I did not understand this value. Is there a unit such as number of cases?

Response 6: Yes, it is observed/expected incidence. This was elaborated at the bottom of the table.

Reviewer 2 Report

The manuscript describes a study on the spatial distribution of FMD in Bangladesh. Spatial and space-time analyses were performed.

The results indicate, that FMD is more common in eastern parts of the country and that pre-monsoon precipitation was positively associated with the occurrence of FMD.

This is a fine geographical work conducted on a dataset that has a lot of weaknesses. However, most of these points are clearly addressed and dealt with in the discussion. The manuscript is well written and elaborated. To me there are no major concerns with this manuscript.

Detailed comments:

Introduction:

“However, there is approximately eight million pigs in Bangladesh [12] and their abundance is more in northern and hilly districts in Bangladesh.” Please correct to “However, there are approximately eight million pigs in Bangladesh [12] and their abundance is more in northern and hilly districts in Bangladesh.”

“Moreover, DLS is doing emergency vaccination (ring vaccination) at outbreak sites [5]” Please describe what DLS is and add a full stop.

Results:

What do you count as “cases”: Animals or farms or outbreaks? Please give a definition and explain how cases are connected with the denominator. Are this only cases in cattle and buffalo or also in goats / sheep?

Discussion

“As all farmers do not have access to upazila veterinary hospitals.” I think there is an error word with “upazila” in the sentence. Please correct. Also in other parts of the manuscript. It is only explained at the end of the discussion but should be explained when it is first mentioned in the manuscript.

Please correct “At present, approximately 3 million doses of trivalent vaccines is being produced for cattle immunization” to “At present, approximately 3 million doses of trivalent vaccines are produced for cattle immunization”. What about other sheep and goats? Please add if there is a vaccine available.

Please check this sentence: “As a result, outbreak occurs in the vaccinated herd also.”

Conclusion

Concerning “The DLS passive surveillance network could be utilized more efficiently if some additional information is included in the report”. I suggest to also include information on the vaccination status of the animals.

Author Response

The manuscript describes a study on the spatial distribution of FMD in Bangladesh. Spatial and space-time analyses were performed.

The results indicate, that FMD is more common in eastern parts of the country and that pre-monsoon precipitation was positively associated with the occurrence of FMD.

This is a fine geographical work conducted on a dataset that has a lot of weaknesses. However, most of these points are clearly addressed and dealt with in the discussion. The manuscript is well written and elaborated. To me there are no major concerns with this manuscript.

Thank you very much for your favorable comments.

Point 1: Introduction:

“However, there is approximately eight million pigs in Bangladesh [12] and their abundance is more in northern and hilly districts in Bangladesh.” Please correct to “However, there are approximately eight million pigs in Bangladesh [12] and their abundance is more in northern and hilly districts in Bangladesh.”

Response 1: Corrected as suggested by the reviewer.

Point 2: Introduction “Moreover, DLS is doing emergency vaccination (ring vaccination) at outbreak sites [5]” Please describe what DLS is and add a full stop.

Response 2: DLS was defined and full stop added.

Point 3: Results

What do you count as “cases”: Animals or farms or outbreaks? Please give a definition and explain how cases are connected with the denominator. Are this only cases in cattle and buffalo or also in goats / sheep?

Response 3: Cases are animals not farms or outbreaks. Case was defined in section 4.1.1. Cumulative incidence was calculated by diving number of cases per district by 10,000 cattle and buffalo population. Yes, only cattle and buffalo cases are included in this study.

Point 4: Discussion

“As all farmers do not have access to upazila veterinary hospitals.” I think there is an error word with “upazila” in the sentence. Please correct. Also in other parts of the manuscript. It is only explained at the end of the discussion but should be explained when it is first mentioned in the manuscript.

Response 4: Upazilla was defined as it mentioned first in the manuscript.

Point 5: Discussion

Please correct “At present, approximately 3 million doses of trivalent vaccines is being produced for cattle immunization” to “At present, approximately 3 million doses of trivalent vaccines are produced for cattle immunization”. What about other sheep and goats? Please add if there is a vaccine available.

Response 5: The sentence was corrected. Sheep and goats are not usually vaccinated.

Point 6: Discussion

Please check this sentence: “As a result, outbreak occurs in the vaccinated herd also.”

Response 6: This sentence was deleted.

Point 7: Conclusion

Concerning “The DLS passive surveillance network could be utilized more efficiently if some additional information is included in the report”. I suggest to also include information on the vaccination status of the animals.

 Response 6: Vaccination status was added.

Reviewer 3 Report

“Foot-and-mouth disease space-time clusters and risk factors in cattle and buffalo in Bangladesh” by Rahman et al describes space-time analysis of reported FMD outbreaks in Bangladesh. The analysis could be beneficial to FMD control in Bangladesh, however there are several points which must be addressed:

Major

Pg2 last paragraph: Why is data presented by district here, but division elsewhere?

How does the proportion of cases in the divisions relate to the cattle and buffalo populations in the divisions?

Pg3 section2.2: was clustering analysis based on cases/100,000 or total cases?

Briefly describe the methods of analysis prior to presenting results since the results section is presented before methods in the manuscript

Rain in pre-monsoon; highest% in post-monsoon

Discuss differences with previous studies. Here you report the highest incidence in the post-monsoon period, but previous studies reported highest incidence in the pre-monsoon period (pg10, 3rd paragraph)

The data collection methods are confusing. It seems that case data is aggregated at the subdistrict level, with only number of cases and dates available, but you report limiting the dataset to cases with characteristic clinical signs. How was this done?

How likely is it that your dataset contains species other than cattle and buffalo?

Please specify that a case is a single animal. Include a discussion of how multiple cases from a single farm would bias your results.

Why was total number of cases and cases/100,000 used in the spatial regression analysis? Using total number of cases would bias results in areas of higher or lower animal density

How did you compare the results from the 3 different cluster analyses to arrive at your presented results?

Minor

Pg2 1st paragraph: start a new paragraph at “A large FMD-susceptible population…”

Pg2 2nd paragraph: write out/define DLS

Pg2 Results: Please specify the years included in the dataset.

Pg3 section 2.2: please define hotspot, high-high, low-low

Pg7 1st paragraph: define EBK

Pg7 section 2.3: define GWR

Pg9 2nd paragraph: define upazila

Figures: increase font size of legends

Supplementary table: include units of measure in the elaboration section

Author Response

Response to Reviewer 3 Comments

“Foot-and-mouth disease space-time clusters and risk factors in cattle and buffalo in Bangladesh” by Rahman et al describes space-time analysis of reported FMD outbreaks in Bangladesh. The analysis could be beneficial to FMD control in Bangladesh, however there are several points which must be addressed

Major

 Point 1:

 Pg2 last paragraph: Why is data presented by district here, but division elsewhere?

Response 1: We used district level FMD case data and analysis was based on cumulative incidence of FMD per 10,000 cattle and buffalo population. Not only here, we presented district level hotspots and clusters also. If some clusters and hotspots were common in one division then we have just shown that division within parenthesis.

Point 2: How does the proportion of cases in the divisions relate to the cattle and buffalo populations in the divisions?

Response 2: Now we have added the average cumulative incidence of FMD per 10,000 population in divisions (Table 1)

Point 3:

Pg3 section2.2: was clustering analysis based on cases/100,000 or total cases?

Response 3: Yes, clustering analysis was based on cases/10,000 population.

Point 4:

Briefly describe the methods of analysis prior to presenting results since the results section is presented before methods in the manuscript

Response 4: It is the journal style to describe results first and methods at the end.

Point 5:

Rain in pre-monsoon; highest% in post-monsoon

Discuss differences with previous studies. Here you report the highest incidence in the post-monsoon period, but previous studies reported highest incidence in the pre-monsoon period (pg10, 3rd paragraph)

Response 5: Table 1 describes the overall proportion of cases (not cumulative incidence per 10,000 population in a district), so it has no spatial component. However, pre-monsoon precipitation was modelled as a spatial risk factor and was significantly associated with district level cumulative incidence of FMD. Another study from Bangladesh also reported the same association. We have now clarified this in the text.

Point 6: The data collection methods are confusing. It seems that case data is aggregated at the subdistrict level, with only number of cases and dates available, but you report limiting the dataset to cases with characteristic clinical signs. How was this done?

Response 6: Yes, cases data is aggregated at sub-district level. The FMD cases were diagnosed by DLS based on the clinical signs mentioned. The case data is aggregated at sub-district level prior to reporting. Thus, we only had access to subdistrict-level aggregated data for analysis.

Point 7:

How likely is it that your dataset contains species other than cattle and buffalo?

Response 7: No, our dataset does not contain FMD of other species like sheep and goats.

Point 8: Please specify that a case is a single animal. Include a discussion of how multiple cases from a single farm would bias your results.

Response: Case was defined as a single animal in section 4.1.1. The data for number of cases per herd was not available. Since cases were aggregated, we argue that there was no systemic bias because we have no reason to suspect that more cases are likely to be diagnosed in some herds (e.g. small herds) that other herds (e.g. large herds). Therefore, between districts there should not be bias caused by this aggregation. We have added details in the discussion.

Point 9:

Why was total number of cases and cases/100,000 used in the spatial regression analysis? Using total number of cases would bias results in areas of higher or lower animal density.

Response 9: There was a mistake in the title of the Table 2, which has now been corrected. We used cases/10,000 for spatial regression analysis i.e. analysis was adjusted for population size.

Point 10: How did you compare the results from the 3 different cluster analyses to arrive at your presented results?.

Response 10: It was based on the similarity between various outputs of the methods.

Minor

Point 1:

Pg2 1st paragraph: start a new paragraph at “A large FMD-susceptible population…”

Response 1: A new paragraph was created from “a large FMD-susceptible population…”

Point 2:

Pg2 2nd paragraph: write out/define DLS

Response 2: DLS was elaborated in 2nd paragraph of page 2.

Point 3:

Pg2 Results: Please specify the years included in the dataset.

Response 3: Years added in result.

Point 4:

Pg3 section 2.2: please define hotspot, high-high, low-low

Response 4: hotspots, high-high and low-low clustered were defined in pages 3 section 2.2.

Point 5:

Pg7 1st paragraph: define EBK

Response 5: EBK was elaborated in 1st paragraph of page 7.

Point 6:

Pg7 section 2.3: define GWR

Response 6: In section 2.3 of page 7 GWR was elaborated.

Point 7:

Pg9 2nd paragraph: define upazila

Response 7: Upazila was defined in second paragraph of page 9.

Point 8:

Figures: increase font size of legends

Response 8: The font size of legends was increased.

Point 9:

Supplementary table: include units of measure in the elaboration section

Response 9: The units of measure were added in the elaboration section of supplementary table.
